# Effects of Temperature on the Tribological Properties of NM600 under Sliding Wear

**DOI:** 10.3390/ma12234009

**Published:** 2019-12-03

**Authors:** Yingchao Pei, Dianxiu Xia, Shouren Wang, Liang Cong, Xuelin Wang, Dongyue Wang

**Affiliations:** 1School of Mechanical Engineering, University of Jinan, Jinan 250022, China; jndxpyc@163.com (Y.P.); me_wangsr@ujn.edu.cn (S.W.); yikuaixiwan666@163.com (D.W.); 2Bisalloy Shangang (Shandong) Steel Plate Co., Ltd., Jinan 250022, China; congliang@bisjigang.com; 3Collaborative Innovation Center of Steel Technology, University of Science and Technology Beijing, Beijing 100000, China; xuelin2076@163.com

**Keywords:** NM600, temperature, sliding wear, wear morphology, wear volume

## Abstract

An investigation on the tribological properties of GCr15 sliding against NM600 was carried out using a high-temperature friction and wear tester. As the temperature rose from room temperature to 300 °C, the average friction coefficient of NM600 increased rapidly, then decreased rapidly, and then became stable. The wear volume and specific wear rate of NM600 increased rapidly, then decreased rapidly, and then increased slowly. The wear mechanism and matrix properties of the tested steel at different temperatures are the main reasons for the above results. At 20–50 °C, the main wear mechanism was adhesive wear, fatigue wear, and abrasive wear. At 100–150 ℃, the wear mechanism was mainly adhesive wear, fatigue wear, abrasive wear, and oxidation wear. At 200–300 °C, the wear mechanism was mainly oxidation wear and abrasive wear.

## 1. Introduction

Low-alloy high-strength wear-resistant steel is widely used in coal mining, construction, metallurgy, and other engineering fields with complex service conditions because of its high hardness and toughness [1,2,3]. At present, the research on the wear process of low-alloy and high-strength wear-resistant steel is mainly focused on the wear at room temperature, though usually the wear of equipment occurs at a certain temperature [4,5,6]; for example, when the middle groove of the fully-mechanized mining scraper conveyor carries out coal mine material transportation. On the one hand, the ambient temperature of the coal roadway is higher than normal temperature. On the other hand, coal, gangue, and other friction with the middle trough also release heat [7,8,9]. Therefore, it is necessary to study the friction and wear properties of low-alloy and high-strength wear-resistant steel at different temperatures. Taking NM600 low-alloy high-strength wear-resistant steel as the research object, through the sliding friction and wear experiments under different temperature conditions, analysis of the influence of temperature on NM600 wear-resistant steel wear behavior rule was completed. The influence of microstructure, matrix properties, and oxidation behavior on the wear behavior of NM600 wear-resistant steel was studied. The wear mechanisms are discussed in this paper in order to improve the service life of the middle trough of the fully-mechanized mining scraper conveyor and provide certain data to support.

## 2. Materials and Methods 

### 2.1. Materials

The experimental material was NM600 low-alloy high-strength wear-resistant steel industrially produced. The chemical composition is shown in Table 1.

The microstructure of the tested steel was mainly lath martensite structure, as shown in Figure 1.

### 2.2. Experimental Methods

The tested steel was processed into wear samples of 10 mm × 10 mm × 5 mm by means of EDM wire cutting. The surface of the samples was gradually polished with sandpaper to #1500, and then the samples were mechanically polished. The final roughness of the sample was about 200 nm. Rtec multifunctional friction and wear tester (San Jose, CA, USA) was used to study the sliding friction and wear behavior between NM600 wear-resistant steel and GCr15 bearing steel at different temperatures. The counter-body was a ball and its diameter was 6.35 mm. Figure 2 shows a schematic diagram of the experimental principle. The temperature was controlled through the high-temperature heating chamber, and the temperature control precision was less than ±10 °C. The experimental parameters were as follows: load, 50 N; amplitude, 4.5 mm; frequency, 4 Hz; and time, 45 min. The experimental temperature ranged from 20 °C to 300 °C. The friction coefficient and the experimental temperature were monitored by the matching program of the testing machine. The profile and volume of wear marks were measured by USP-Sigma white-light interferometer (San Jose, CA, USA). The microscopic wear morphology was observed by TESCAN-MIRA3 scanning electron microscope (HITACHI, Tokyo, Japan). Before and after each experiment, the samples were ultrasonic cleaned and then blown dried.

## 3. Results

### 3.1. Effects of Temperature on Friction Coefficient

As can be seen from Figure 3, at the initial stage of wear, the friction coefficient of the tested steel fluctuated greatly at various temperatures. With the continuous wear time, the friction coefficient of the tested steel gradually tended to be stable. The difference is that when the experimental temperature was 50 °C, the friction coefficient of the tested steel needed a long time to reach a stable state; while, at the other experimental temperatures, the friction coefficient of the tested steel all reached a stable state in a short time.

As can be seen from Figure 4, when the experimental temperature increased from 20 °C to 50 °C, the average friction coefficient of the tested steel increased rapidly. When the experimental temperature continued to rise from 50 °C to 200 °C, the average friction coefficient of the tested steel decreased rapidly. When the experimental temperature continued to rise from 200 °C to 300 °C, the average friction coefficient of the tested steel basically remained stable. It can be seen that when the experimental temperature exceeded 200 °C, the average friction coefficient of the tested steel was less sensitive to temperature, indicating that the wear mechanism of the tested steel may have changed.

### 3.2. Effects of Temperature on Macroscopic Wear Morphology and Wear Volume

As shown in Figure 5 and Figure 6, when the experimental temperature rose from 20 °C to 50 °C, the wear degree of the tested steel increased rapidly. When the experimental temperature continued to rise from 50 °C to 150 °C, the wear degree of the tested steel decreased rapidly. When the experimental temperature increased from 150 °C to 300 °C, the wear degree of the tested steel increased slowly.

The calculation formula of specific wear rate is as follows [10]:*W_R_*= *W_V_*/(*F_H_ S*)(1)
where *W_R_* is the specific wear rate, *W_V_* is the wear volume, *F_H_* is the force, and *S* is the wear length.

As shown in Figure 7 and Figure 8, when the experimental temperature increased from 20 °C to 50 °C, the wear volume and specific wear rate of the tested steel increased rapidly. When the experimental temperature increased from 50 °C to 150 °C, the wear volume and specific wear rate of the tested steel decreased rapidly. When the experimental temperature increased from 150 °C to 300 °C, the wear volume and specific wear rate of the tested steel increased slowly.

In conclusion, with the increase of the experimental temperature, the macroscopic wear morphology, wear volume, and specific wear rate of the tested steel all increased rapidly first, then decreased rapidly, and then increased slowly. There are two experimental temperatures worth noting. First, when the experimental temperature was 50 °C, the wear of the tested steel was the most serious, and the wear degree of the tested steel also changed from a rapid increase to a rapid decrease at this temperature point. Second, when the experimental temperature was 150 °C, the wear degree of the tested steel was the least, and the wear degree of the tested steel also changed from a rapid decrease to a slow increase at this temperature point. It can be concluded that when the experimental temperature was 50 °C and 150 °C, the wear mechanism of the tested steel may have changed.

### 3.3. Effects of Temperature on Microscopic Wear Morphology

As shown in Figure 9, when the experimental temperature was 20 °C, 50 °C, 100 °C, and 150 °C, the wear surface of the tested steel showed different degrees of fatigue spalling, microcutting, and plastic deformation. When the experimental temperature was 200 °C, 250 °C, and 300 °C, the worn surface of the tested steel mainly showed grooves, plastic deformation, and a little fatigue peeling. It is not difficult to find that when the experimental temperature was 50 °C, the fatigue spalling phenomenon of the worn surface of the tested steel was the most serious. However, when the experimental temperature exceeded 50 °C, the fatigue spalling of the worn surface of the tested steel decreased with the increase of the experimental temperature. By observing the wear morphology of high-power SEM, it was found that when the experimental temperature was 20 °C and 50 °C, a large amount of loose abrasive particles had accumulated in the fatigue spalling pit of the worn surface of the tested steel. When the experimental temperature was 150 °C, only a small amount of scattered abrasive particles were found in the fatigue spalling pit of the worn surface of the tested steel, and most of the scattered abrasive particles were sintered together to form a dense glaze layer. When the experimental temperature was 200 °C, 250 °C, and 300 °C, with the increase of the experimental temperature, the glaze layer on the worn surface of the tested steel gradually cracked and the amount of scattered abrasive particles gradually increased.

The EDS analysis results (Figure 10 and Table 2) showed that the oxygen content of both the worn surface and the unworn surface increased gradually with the increase of the experimental temperature. When the experimental temperature was 20 °C and 50 °C, the oxygen content of the worn surface was similar to that of the unworn surface, indicating that there was almost no oxidation wear at this temperature. When the experimental temperature was 100–300 °C, the difference of oxygen content between the worn surface and the unworn surface gradually increased, indicating that oxidation wear occurred in the process of wear, and the degree of oxidation wear gradually intensified.

## 4. Discussion

Studies have shown that the friction coefficient mainly depends on the contact state between the contact surfaces of the friction pairs [11,12]. In this experiment, with the increase of the experimental temperature, the average friction coefficient of the tested steel increased first, then decreased, and then tended to be stable. Combined with the analysis of the micro-wear morphology (Figure 9) and the EDS analysis results (Figure 10 and Table 2) of the tested steel at different temperatures, when the experimental temperature increased from 20 °C to 50 °C, the fatigue spalling area of the worn surface of the tested steel increased significantly, and a large amount of abrasive residue on the worn surface of the tested steel resulted in the increase of friction coefficient. However, when the experimental temperature exceeded 50 °C, the fatigue spalling area of the worn surface of the tested steel was gradually reduced. Especially, when the experimental temperature rose to 200 °C, 250 °C, and 300 °C, the majority of the worn surface of the tested steel was covered by the enamel layer formed by the sintering of the loose abrasive particles, which made the friction coefficient of the tested steel decrease rapidly. It then tended to be stable.

The changing trend of wear volume and specific wear rate obtained from the experiment can directly show the wear degree of the tested steel at different temperatures. It was found that there are two key transition temperatures for the trend change of wear degree of the tested steel: 50 °C and 150 °C, which should be related to the change of wear mechanism of the tested steel. By observing and analyzing the micro-wear morphology of the tested steel at different temperatures, it can be concluded that when the experimental temperature was 20 °C and 50 °C, the wear mechanism of the tested steel was mainly manifested as adhesive wear, fatigue wear, and abrasive wear of different degrees. Among them, when the experimental temperature was 20 °C, under the action of external load, the abrasion contact surface between NM600 tested steel and GCr15 bearing steel was stuck, and material transfer and spalling occurred between the two contact surfaces in the subsequent sliding friction process. In addition, the continuous reciprocating sliding friction movement caused fatigue spalling of the wear contact surface under the action of reciprocating alternating contact stress. Adhesive wear and fatigue wear produced a certain amount of abrasive particles, which lead to abrasive wear. When the experiment temperature was 50 °C, the worn surface of the tested steel adhesive wear and fatigue wear also existed. However, with the increase of the experimental temperature, the amount of abrasive particles caused by adhesive wear and fatigue wear increased, which aggravates the abrasive wear of the experimental steel and eventually leads to serious wear of the tested steel [13,14,15,16]. When the experimental temperature rose to 100 °C and 150 °C, the wear mechanism of the tested steel was mainly manifested as adhesive wear, fatigue wear, abrasive wear, and oxidation wear of different degrees. With the increase of the experimental temperature, the abrasive particles generated by adhesive wear and fatigue wear were gradually oxidized and sintered under the action of high temperature to form a stable and dense enamel layer structure, which can effectively reduce the direct contact between the wear surfaces, thus playing a role in protecting the matrix and reducing wear [17,18,19]. When the experimental temperature rose to 200 °C, 250 °C, and 300 °C, the wear mechanism of the tested steel was mainly oxidation wear and abrasive wear. However, the oxidation wear mentioned here is substantially different from the oxidation wear mentioned above at 100 ℃ and 150 °C. The oxidation wear mentioned here is called severe oxidation wear, while the former is called slight oxidation wear [20,21]. When the experiment temperature rose to 200 °C, 250 °C, and 300 °C, while the tested steel wear surface residual wear debris can also be formed under the action of high-temperature sintering of the enamel layer, continuous high temperature in the process of wear and tear effect made grain coarsening, transformation, recrystallization, and grain size response organization transformation inevitable in the tested steel. This led to the decrease of the tested steel substrate softening and strength. Losing the effective support matrix form of the enamel layer cannot make stability exist in the worn surface of the tested steel at the same time because of the effect of external load making the enamel layer of the worn surface of the tested steel quickly break and flake, thus unable to effectively slow down the effect of matrix wear, unlike in repeated sintering where it peels off. The sintering process leads to loss of substrate material due to the increasing degree of tested steel wear again [22,23,24,25].

The wear test results at different temperatures are summarized as shown in Table 3.

## 5. Conclusions

When the experimental temperature was 50 °C, the increase of stray abrasive particles on the worn surface of the tested steel was the main reason leading to the increase of the average friction coefficient. When the experimental temperature exceeded 50 °C, the average friction coefficient of the tested steel decreased mainly because smooth enamel layer gradually formed on the worn surface of the tested steel.From 20 °C to 300 °C, the wear mechanism of the tested steel changed twice at 50 °C and 150 °C, respectively. From 20 °C to 50 °C, the wear mechanism of the tested steel was mainly adhesive wear, fatigue wear, and abrasive wear. From 100 °C to 150 °C, the wear mechanism of the tested steel mainly included fatigue wear, abrasive wear, and oxidation wear. From 200 °C to 300 °C, the wear mechanism of the tested steel was mainly oxidation wear and abrasive wear.When the experimental temperature was 100 °C and 150 °C, the oxidation wear of the worn surface of the tested steel was mild oxidation wear, which can effectively protect the substrate and slow down the wear. When the experimental temperature was 200 °C, 250 °C, and 300 °C, the oxidation wear of the worn surface of the tested steel was serious oxidation wear, which cannot protect the substrate and slow down the wear.

## Figures and Tables

**Figure 1 materials-12-04009-f001:**
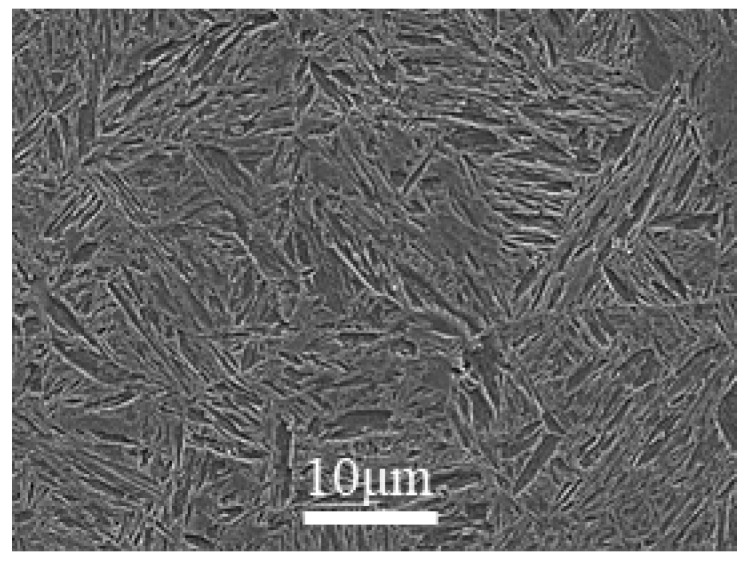
SEM microstructure of NM600.

**Figure 2 materials-12-04009-f002:**
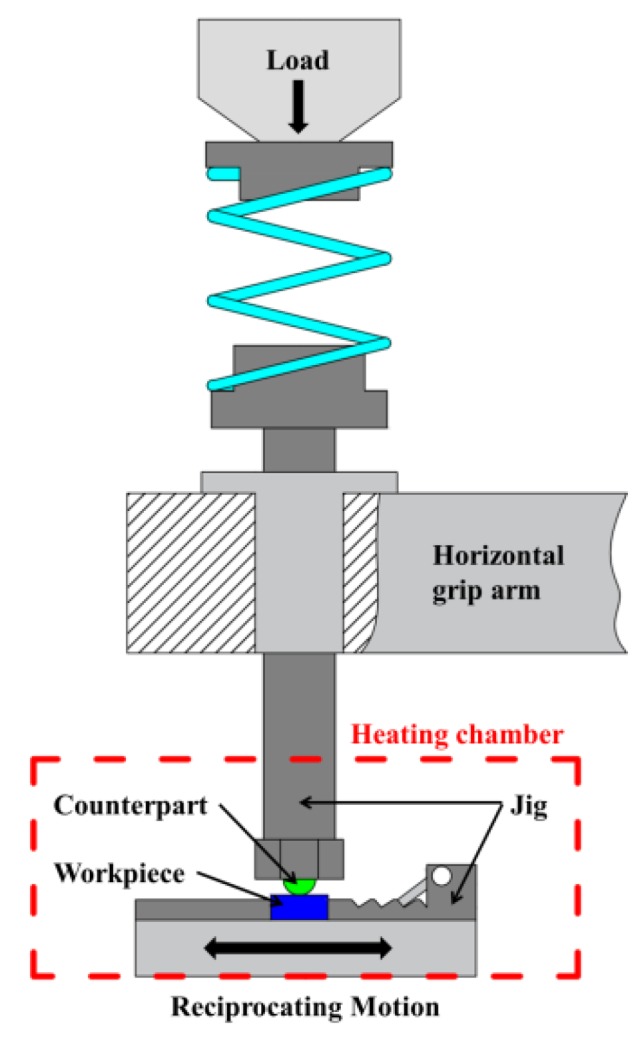
Principle diagram of high-temperature sliding wear.

**Figure 3 materials-12-04009-f003:**
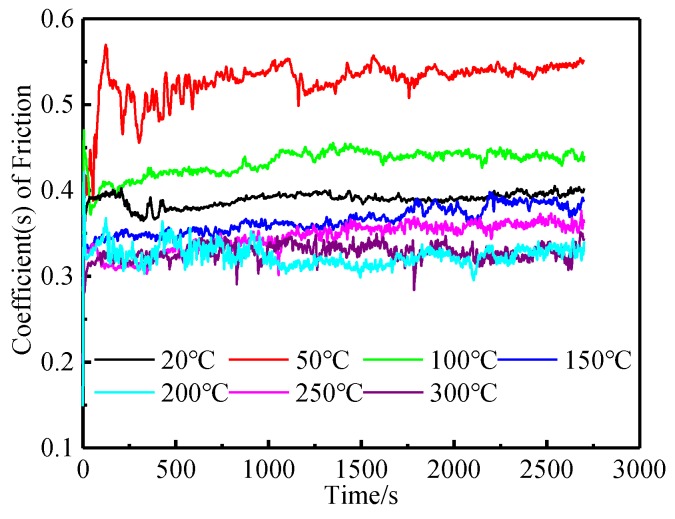
Friction coefficient versus time at different temperatures.

**Figure 4 materials-12-04009-f004:**
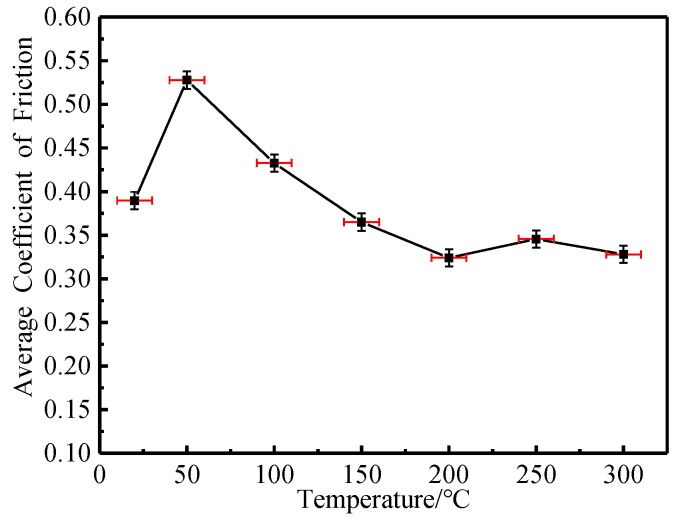
Average friction coefficient versus temperature.

**Figure 5 materials-12-04009-f005:**
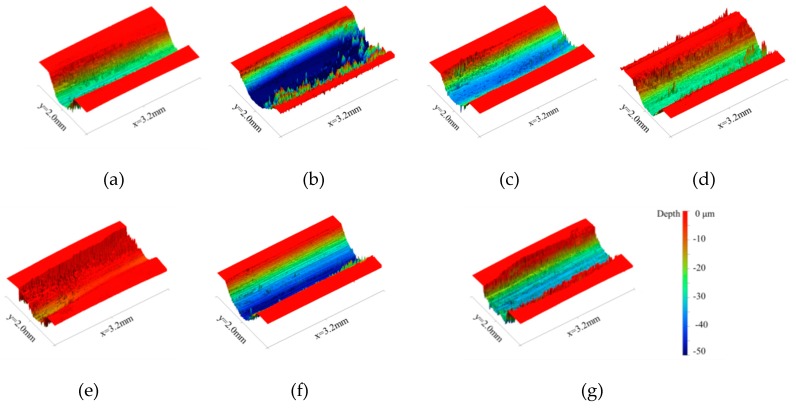
Three-dimensional wear morphology at different temperatures. (**a**) 20 °C. (**b**) 50 °C. (**c**) 100 °C. (**d**) 150 °C. (**e**) 200 °C. (**f**) 250 °C. (**g**) 300 °C.

**Figure 6 materials-12-04009-f006:**
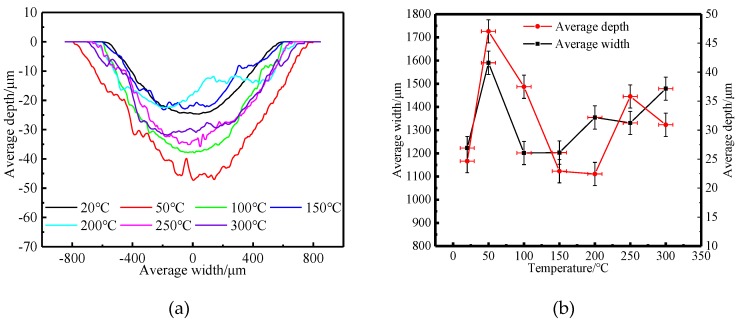
Two-dimensional wear morphology versus temperature. (**a**) Grinding mark section profile at different temperature. (**b**) Average wear depth and width at different temperature.

**Figure 7 materials-12-04009-f007:**
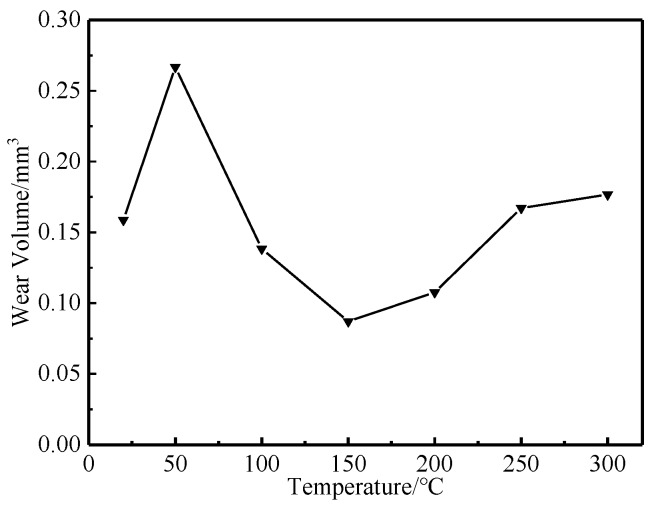
The wear volume (the whole wear length) versus temperature.

**Figure 8 materials-12-04009-f008:**
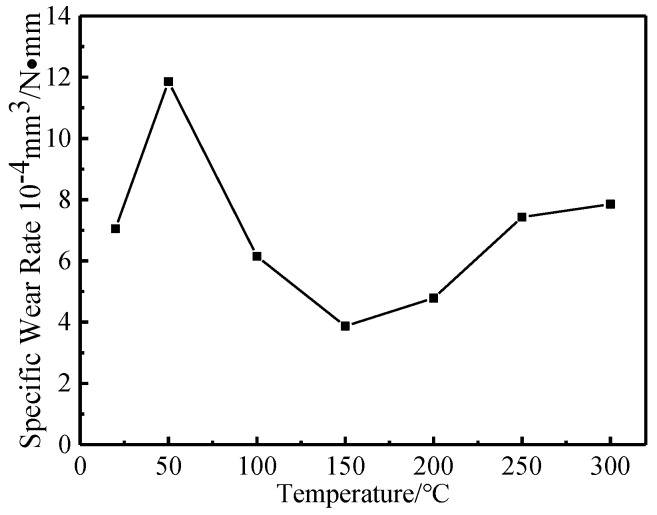
Specific wear rate (the whole wear length) versus temperature.

**Figure 9 materials-12-04009-f009:**
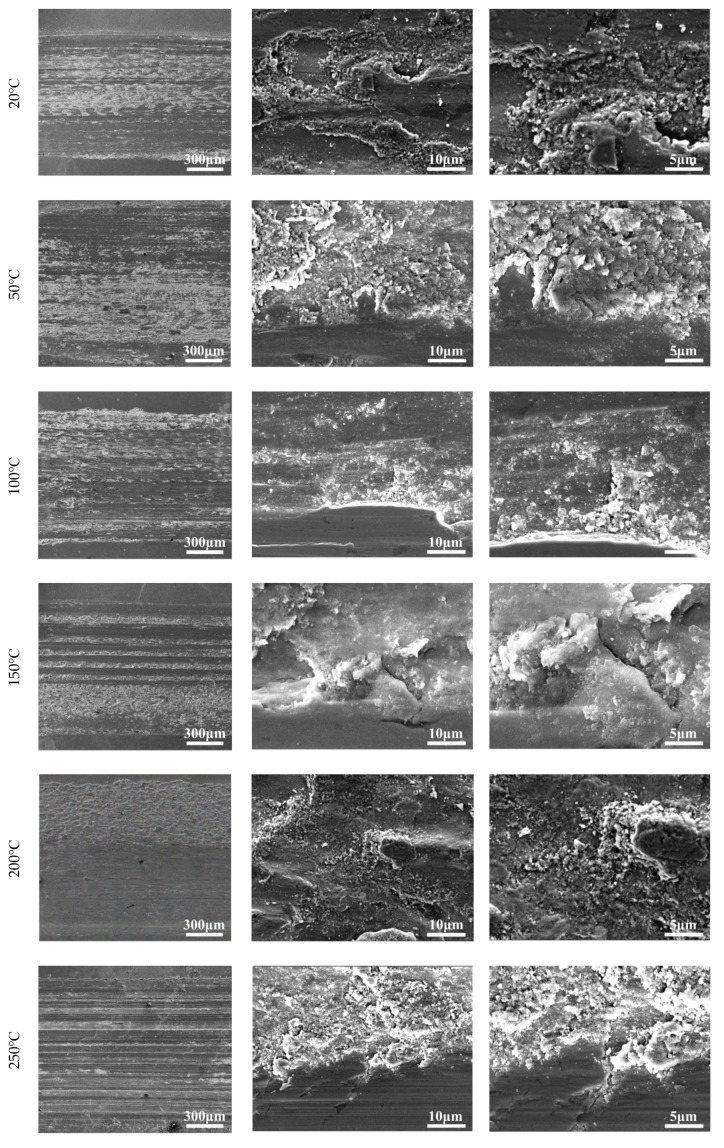
Microscopic wear morphology at different temperatures (SEM).

**Figure 10 materials-12-04009-f010:**
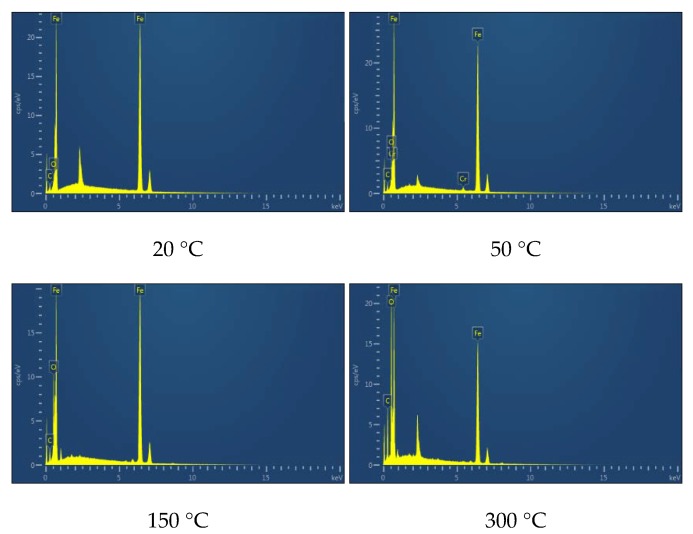
EDS spectra of worn surfaces at different temperatures.

**Table 1 materials-12-04009-t001:** The chemical composition of NM600 (mass fraction, %).

w(C)%	w(Si)%	w(Mn)%	w(P)%	w(S)%	w(Cr)%	w(Ni)%	w(B)%	w(Mo)%
0.44–0.46	0.34–0.36	0.38–0.42	≤0.02	≤0.005	1.1–1.3	0.9–1.1	0.002	0.28–0.32

**Table 2 materials-12-04009-t002:** EDS result (oxygen content) of worn and unworn surfaces at different temperatures (at%).

	20 °C	50 °C	100 °C	150 °C	200 °C	250 °C	300 °C
**Worn Surfaces**	3.3	6.6	12.3	18.7	23.6	27.3	33.4
**Unworn Surfaces**	3.1	6.2	10.1	13.2	17.7	21.5	25.3

**Table 3 materials-12-04009-t003:** Summary of wear test results at different temperatures.

	20 °C	50 °C	100 °C	150 °C	200 °C	250 °C	300 °C
Average Friction Coefficient (±0.001)	0.3896	0.5277	0.4327	0.3650	0.3241	0.3456	0.3281
Wear Volume/mm^3^ (±0.001)	0.1587	0.2668	0.1384	0.0871	0.1077	0.1672	0.1767
Specific Wear Rate 10^−4^ mm^3^/mm	7.0533	11.8578	6.1511	3.8720	4.7867	7.4311	7.8533
Adhesive Wear	√	√					
Fatigue Wear	√	√	√	√			
Abrasive Wear	√	√	√	√	√	√	√
Slight Oxidation Wear			√	√			
Severe Oxidation Wear					√	√	√

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
