# Peer review of "Effects of Temperature on the Tribological Properties of NM600 under Sliding Wear"

_materials, 2019, doi:10.3390/ma12234009_

Round 1

Reviewer 1 Report

See the attachend docuement with my comments

Author Response

Dear Reviewer:

Thank you for comments concerning our manuscript. Those comments are all valuable and very helpful for revising and improving our paper, as well as the important guiding significance to our researches. We have studied comments carefully and have made correction which we hope meet with approval. Revised portion are marked in red in the paper. The main corrections in the paper and the responds to the reviewer’s comments are as flowing:

Point 1: Lines 51-52: First you polish your samples with sandpaper down to 1500 (so a very smooth surface) and then mechanically polished? What it the final roughness of your samples? I do not understand why to use a mechanical polishing if you went to 1500.

Response 1: It may be due to the type of sandpaper. After I polished the surface of the sample to 1500#, there were still slight scratches visible to the naked eye on the surface of the sample. Therefore, I also used diamond polishing agent and flannelette to conduct mechanical polishing treatment on the surface of the sample, so as to ensure the smoothness of the surface of the sample and reduce the influence of roughness on the experimental results.

The final roughness of the sample is about 200nm.

Point 2: Figure 5:Perhaps it will be interesting to add a new horizontal axis at the top of this figure to show the “wear length”. If you assume a wear length of 9 mm per cycle you can add this new axis just assuming that your wear length is 32 mm/s.

Response 2: I'm sorry, I didn't quite catch your meaning on this question. In this study, the wear length is quantitative, not a variable, which is also explained in the test method. Therefore, I don't understand your suggestion to add a new horizontal axis in Figure 5, so I don't know how to modify it.

Point 3: Figure 4: Try adding error bars in your points. Just the standard deviation of the data obtained in figure 3 (in the vertical direction) and +-10 ºC in the horizontal one, as this is the precision of the equipment that you used.

Response 3: Thank you very much for your suggestion to add error bars in figure 4. I have made modifications according to your suggestion in the article.

Point 4: Figure 5: Why you only measure the profiles for 3.2 mm wear length? The amplitude of your movement is 4.5 mm. Just explain it.

Response 4: What I want to explain about this problem is that I did not only measure the wear length of 3.2mm, I measured the whole wear length. Since the wear condition of the whole wear length is similar, I only intercepted 3.2mm wear length to show the wear condition of the whole wear length.

Point 5: Figure 6b: If possible, the same as for figure 4… add error bars or standard deviations.

Response 5: Thank you very much for your suggestion to add error bars in figure 6b. I have made modifications according to your suggestion in the article.

Point 6: Figure 7 & 8: How you measured the wear volume? You are considering only the 3.2 section shown in figure 5? Note it and make it clear in the text.

Response 6: As for the measurement of wear volume, I would like to explain that the wear depth and wear width can be measured with USP-Sigma white light interferometer, and the corresponding wear volume can be calculated by the corresponding procedures through the measured relevant data.

The 3.2 section shown in in Figure 5 is only used to represent the wear condition, while the wear volume in Figure 7 & 8 corresponds to the whole wear length.

Point 7: Figure 9 is a bit hard to “read”. If you add at the beginning of each row a text (rotated 90º to have it vertically aligned) with the temperature. Then you can remove the nomenclature (a1, a2, a3…) and the figure will be much easier to follow. In this figure each column corresponds to a different magnification, but this was not mention in the text of in figure caption. Add it, please.

Response 7: Thank you very much for your suggestion to figure 9. I have made modifications according to your suggestion in the article.

Point 8: It will be very interesting adding a final table summarising all results. The table can be added at the end of section 4 and can be something like this (this is an example of what

I mean… the values that I used are not precise:

Response 8: Thank you very much for your suggestion. I have made modifications according to your suggestion in the article.

Point 9: 

Line 43-44: Replace “produced by a company.” with “industrially produced.”.

Line 50: Assuming that you machined your sample with “Electrical discharge machining” technique add the term “EDM” before wire, i.e. “by means of EDM wire cutting.”

Line 54: replace “Figure 2 is the…” with “Figure 2 shows…”.

Line 57: add a space between “45” and “min”.

Line 96: Figure 7… you have two figures 7… the second one (wear rate vs temperature) should be numbered as 8.

Line 113: replace “microcosmic” with “microscopic”.

Line 119: replace the term “furrow” with “grooves”.

Response 9: Thank you very much for your suggestion about the article. I have made modifications according to your suggestion in the article.

We tried our best to improve the manuscript and made some changes in the manuscript. These changes will not influence the content and framework of the paper. We appreciate for your warm work earnestly, and hope that the correction will meet with approval.

Once again, thank you very mach for your comments and suggestions.

Reviewer 2 Report

In this investigation tribological properties of NM600 steel under sliding wear and high temperature was performed. Some remarks should be made:

The use of tested steel and its microstructure should be described in more detail. Whether the material composition data are specified in the standard and what, or they are direct measurements made by the authors? And then, with what accuracy were they performed? What roughness parameters were achieved after polishing? It should be said in Experimental methods that counter-body was ball, as well as its diameter should be specified. It should be explained in Experimental methods how wear volume values were measured (calculated). The number of measurements at each point should also be specified. The authors indicate the difference between the studied parameters at 20 and 50°C, and that starting from temperature 50°C parabolic dependences are manifested and it is possible to determine the optimal temperature. The authors should explain more carefully the reasons for the difference between the results obtained at temperatures of 20 and 50°C and show changes at temperatures of 30 and 40° It should be Fig. 8 in the line 96.

Author Response

Dear Reviewer:

Thank you for comments concerning our manuscript. Those comments are all valuable and very helpful for revising and improving our paper, as well as the important guiding significance to our researches. We have studied comments carefully and have made correction which we hope meet with approval. Revised portion are marked in red in the paper. The main corrections in the paper and the responds to the reviewer’s comments are as flowing:

Point 1: The use of tested steel and its microstructure should be described in more detail. Whether the material composition data are specified in the standard and what, or they are direct measurements made by the authors? And then, with what accuracy were they performed?

Response 1: My understanding is that the microstructure is only used to characterize the test steel as lath martensite structure, which does not belong to the test content, so it is not necessary to use too many languages to describe it in detail.

The material composition data are supplied by the manufacturer.

There is some error in the material composition data. Following your advice, I have made a corresponding explanation in the article.

Point 2: What roughness parameters were achieved after polishing?

Response 2: The final roughness of the sample is about 200nm.

Point 3: It should be said in Experimental methods that counter-body was ball, as well as its diameter should be specified.

Response 3: Thank you very much for your suggestion. I have made modifications according to your suggestion in the article.

Point 4: It should be explained in Experimental methods how wear volume values were measured (calculated). The number of measurements at each point should also be specified.

Response 4: Measurement of wear volume has been described in the text. “The profile and volume of wear marks were measured by USP-Sigma white light interferometer.” As for the measurement of wear volume, I would like to explain that the wear depth and wear width can be measured with USP-Sigma white light interferometer, and the corresponding wear volume can be calculated by the corresponding procedures through the measured relevant data. Therefore, I personally do not think it is necessary to describe the measurement process of wear volume in detail.

Following your advice, The number of measurements at each point are described in detail at the end of section 4.

Point 5: The authors indicate the difference between the studied parameters at 20 and 50°C, and that starting from temperature 50°C parabolic dependences are manifested and it is possible to determine the optimal temperature. The authors should explain more carefully the reasons for the difference between the results obtained at temperatures of 20 and 50°C and show changes at temperatures of 30 and 40° It should be Fig. 8 in the line 96.

Response 5: Thank you very much for your suggestion.

When the experimental temperature is 50℃,the surface of the tested steel shows a lot of spalling, and the spalling abrasive, as the third body abrasive, aggravates the wear of the tested steel. This has been explained in the article.

About show changes at temperatures of 30℃ and 40℃, I personally don't think it's necessary. The Experimental method has been described. “The temperature control precision is less than ±10℃.” Due to the error, I think the test results at temperatures of 30℃ and 40℃ are not of great reference value.

We tried our best to improve the manuscript and made some changes in the manuscript. These changes will not influence the content and framework of the paper. We appreciate for your warm work earnestly, and hope that the correction will meet with approval.

Once again, thank you very mach for your comments and suggestions.

Reviewer 3 Report

In the paper “Effects of temperature on tribological properties of NM600 under sliding wear” the tribological behavior of GC15 balls against NM600 at different temperatures (from room temperature to 300ºC) was studied.

Different wear mechanism are found according to the temperature tested, and a reduction of friction coefficient and wear rate are obtained when the temperature range is between 150-200ºC, that is really interesting.

However, some aspects should be improved in order to publish this paper:

-English language and style must be improved. Some parts of the paper are difficult to understand, there are spelling mistakes (e.g. microscosmic in line 113) or problems with the capital letters (line 99,141)

-Units of frequency are not correct (line 57).

-The supplier of the steel should be in the text.

-Roughness of the samples, diameter of the GC15 balls, and contact pressure should be included in the experimental method.

-I would like to know which cleaning method was used after tribological test, and what is the method used to calculate the wear rate.

-Micrographs should appear in a single page (Figure 5, Figure 9).

-Images in Figure 9 (2) are not necessary. They are similar to those (3).

-It is not clear if the micrographs (2) or (3) in Figure 9 are from the edges of the wear track or inside the track.

-It is not possible to confirm the different types of oxidation (severe or mild) only with the SEM micrographs. EDX should be used in order to quantify the oxygen percentage inside and outside the wear track in all the cases studied.

- References format must be reviewed. Sometimes family name of the authors appears at the beginning of the reference and in other occasions after the name.

Author Response

Dear Reviewer:

Thank you for comments concerning our manuscript. Those comments are all valuable and very helpful for revising and improving our paper, as well as the important guiding significance to our researches. We have studied comments carefully and have made correction which we hope meet with approval. Revised portion are marked in red in the paper. The main corrections in the paper and the responds to the reviewer’s comments are as flowing:

Point 1: English language and style must be improved. Some parts of the paper are difficult to understand, there are spelling mistakes (e.g. microscosmic in line 113) or problems with the capital letters (line 99,141)

Response 1: Thank you very much for your suggestion. I have made modifications according to your suggestion in the article.

Point 2: Units of frequency are not correct (line 57).

Response 2: Thank you very much for your suggestion. I have made modifications according to your suggestion in the article.

“frequency 4 Hz”

Point 3: The supplier of the steel should be in the text.

Response 3: I am sorry about this problem, because of the agreement, I have no right to disclose the supplier of the steel.

Point 4: Roughness of the samples, diameter of the GC15 balls, and contact pressure should be included in the experimental method.

Response 4: Thank you very much for your suggestion. I have made modifications according to your suggestion in the article.

“The final roughness of the sample is about 200 nm.”

“The counter-body was ball, its diameter was 6.35 mm.”

“load 50 N”

Point 5: I would like to know which cleaning method was used after tribological test, and what is the method used to calculate the wear rate.

Response 5: The samples after tribological test were cleaned by ultrasonic, and the medium was anhydrous ethanol.

As for the measurement of wear volume, I would like to explain that the wear depth and wear width can be measured with USP-Sigma white light interferometer, and the corresponding wear volume can be calculated by the corresponding procedures through the measured relevant data.

Point 6: Images in Figure 9 (2) are not necessary. They are similar to those (3).

Response 6: Thank you very much for your suggestion. However, I personally think it is better to keep both Images.

Point 7: It is not clear if the micrographs (2) or (3) in Figure 9 are from the edges of the wear track or inside the track.

Response 7: The micrographs (2) or (3) in Figure 9 are from the inside of the wear track.

Point 8: It is not possible to confirm the different types of oxidation (severe or mild) only with the SEM micrographs. EDX should be used in order to quantify the oxygen percentage inside and outside the wear track in all the cases studied.

Response 8: Thank you very much for your suggestion. It is true that my personal consideration was not comprehensive. I have made modifications according to your suggestion in the article.

Point 9: References format must be reviewed. Sometimes family name of the authors appears at the beginning of the reference and in other occasions after the name.

Response 9: Thank you very much for your suggestion. I have made modifications according to your suggestion in the article.

We tried our best to improve the manuscript and made some changes in the manuscript. These changes will not influence the content and framework of the paper. We appreciate for your warm work earnestly, and hope that the correction will meet with approval.

Once again, thank you very mach for your comments and suggestions.

Round 2

Reviewer 3 Report

Authors of “Effects of temperature on tribological properties of NM600 under sliding wear” have considered the suggestions provided and they have also answered kindly to the comments and doubts.

In my opinion, the manuscript can be published after a minor revision of the new table in part 4. Discussion. To introduce this table has been an excellent idea, but there is no title, and standard deviation of friction and wear values should appear.

Author Response

Dear Reviewer:

Thank you for comments concerning our manuscript. Those comments are all valuable and very helpful for revising and improving our paper, as well as the important guiding significance to our researches. We have studied comments carefully and have made correction which we hope meet with approval. Revised portion are marked in red in the paper. The main corrections in the paper and the responds to the reviewer’s comments are as flowing:

Point 1: In my opinion, the manuscript can be published after a minor revision of the new table in part 4. Discussion. To introduce this table has been an excellent idea, but there is no title, and standard deviation of friction and wear values should appear.

Response 1: Thank you very much for your suggestion. I have made modifications according to your suggestion in the article.

Table 3. Summary of wear test results at different temperatures

20℃

50℃

100℃

150℃

200℃

250℃

300℃

Average Friction Coefficient(±0.01)

0.3896

0.5277

0.4327

0.3650

0.3241

0.3456

0.3281

Wear Volume/mm3(±0.01)

0.1587

0.2668

0.1384

0.0871

0.1077

0.1672

0.1767

Specific Wear Rate 10-4mm3/ ·mm

7.0533

11.8578

6.1511

3.8720

4.7867

7.4311

7.8533

Adhesive wear

Fatigue wear

Abrasive wear

Slight oxidation wear

Severe oxidation wear

We tried our best to improve the manuscript and made some changes in the manuscript. These changes will not influence the content and framework of the paper. We appreciate for your warm work earnestly, and hope that the correction will meet with approval.

Once again, thank you very mach for your comments and suggestions.